# Experiences shaping research career intention among Black, Hispanic, and Indigenous-identifying first-year allopathic medical students in the United States: A qualitative study

Shruthi Venkataraman[1]*, Meghan O'Connell[2], Adeola Ayedun[1], Allison Aviles[2], Alexandra M. Hajduk[2], Mytien Nguyen[3], Gbenga Ogedegbe[4], Laura Castillo-Page[5], David Henderson[6], Judee Richardson[7], Leslie A. Curry[8], John Paul Sánchez[9], Rachel K. Wolfson[10,11], Sarwat I. Chaudhry[2], Dowin Boatright[1]

**1** Ronald O. Perelman Department of Emergency Medicine, NYU Grossman School of Medicine, New York, New York, United States of America, **2** Department of Internal Medicine, Yale School of Medicine, New Haven, Connecticut, United States of America, **3** Department of Immunobiology, Yale School of Medicine, New Haven, Connecticut, United States of America, **4** Institute for Excellence in Health Equity, New York University Grossman School of Medicine, New York, New York, United States of America, **5** American Board of Medical Specialties, Chicago, Illinois, United States of America, **6** Department of Family Medicine, University of Connecticut, Storrs, Connecticut, United States of America, **7** Department of Medical Education, College of Medicine, University of Illinois, Chicago, Illinois, United States of America, **8** Department of Health Policy, Yale School of Public Health, New Haven, Connecticut, United States of America, **9** Department of Emergency Medicine, Albert Einstein College of Medicine, New York, New York, United States of America, **10** Pritzker School of Medicine, University of Chicago, Chicago, Illinois, United States of America, **11** Department of Pediatrics, Pritzker School of Medicine, University of Chicago, Chicago, Illinois, United States of America

\* shruthi.venkataraman@nyulangone.org

## Abstract

### Objective

To examine the early experiences influencing research career intentions (RCI) among MD students from racial and ethnic backgrounds underrepresented in medicine (URiM).

### Methods

We conducted semi-structured, in-depth interviews with 31 first-year URiM medical students from MD-granting programs across the US to examine student-reported experiences influencing RCI.

### Results

Participants were first-year medical students (N = 31; mean age 24.8 ± 2.6 years; 64.5% female) identifying as Black (38.7%), Hispanic (32.3%), or Multiracial (29%). Four themes were identified: (1) structured premedical research exposure was described as pivotal to developing early research engagement and interest in research careers; (2) research orientations reflected a commitment to using research as a vehicle for social justice and community impact; (3) high-quality research

**Data availability statement:** Due to the highly sensitive nature of the data collected in this study, the data cannot be made publicly available in order to protect participant confidentiality and to comply with the terms of the verbal informed consent obtained from participants. At the time of consent, participants were informed that access to their information would be restricted to the research team and individuals responsible for research oversight, and that the data would not be shared with any parties outside the investigative team, including program leadership at their respective institutions. Questions regarding data access or participant protections may be directed to the Yale University Human Research Protection Program (HRPP@yale.edu).

**Funding:** This study was funded by the National Institute of Health (NIH), project number R35GM153263. The NIH did not have any role in the study outside of funding. The authors listed are all independent from the NIH and had full access to all the data in the study and can take full responsibility for the integrity of the data and the accuracy of the data analysis.

**Competing interests:** The authors have declared that no competing interests exist.

mentorship was characterized by authentic relational investment, skill development, and the distinct value of racial and ethnic identity-concordant role models; and (4) the research arms race for residency placement was described as amplifying systemic inequities that constrained students' research engagement. Across themes, students described tensions between academic research culture and their personal values, including a desire to advance equity and contribute meaningfully to science. For some, this misalignment made research feel less purposeful or personally aligned.

## Conclusions

Medical training programs seeking to support URiM students' RCI should invest in structured premedical research programs and expand access to research mentorship that is both relationally invested and identity concordant. Efforts to cultivate sustained engagement should address publication pressures tied to residency competitiveness, which amplify structural barriers and misalign with students' motivations for pursuing research. Broadening definitions of scholarly contribution and fostering research environments that affirm students' values may be critical to building a robust physician-scientist workforce.

## Introduction

The United States faces a persistent shortage of physician-scientists (physicians who conduct research). Only about 1.5% of U.S. physicians are engaged in research, and even fewer identify as members of racial and ethnic groups underrepresented in medicine (URiM), including Black or African American, Hispanic or Latinx, American Indian or Alaska Native (AIAN), and Native Hawaiian or other Pacific Islander (NHPI) [1]. A 2014 National Institutes of Health (NIH) workforce report highlighted this gap, noting that physician-scientist trainees from URiM backgrounds have stagnated in number and comprised only ~7% of NIH research grant applicants and 4.7% of awardees [1]. Expanding the pool of physician-scientists, especially with individuals from URiM groups, is crucial for scientific advancement. Diverse research teams generate higher-impact science, marked by greater innovation, broader collaboration, and findings that are more generalizable across populations [2–6]. For these reasons major funding and training organizations have prioritized diversifying the physician-scientist pipeline [7].

Medical school is a period of evolution in career intentions, including plans to pursue physician-scientist career pathways [8–11]. However, research experiences among URiM and non-URiM trainees differ [12–14]. URiM medical students are less likely to develop and more likely to lose intent to pursue academic careers (settings where physicians most often conduct research) compared with their non-URiM peers [10]. Emerging evidence suggests they are also less likely to espouse research career intention (RCI) [7]. Consistent with this, our unpublished analyses of contemporary national data show that URiM students enter medical school with similar RCI levels as their peers but are less likely to maintain them by graduation. The conditions underlying this divergence remain poorly understood.

Research exposure in the premedical years and in the first year of medical school may be salient in how RCI begins to form [15–17], and may provide a lens through which later research experiences are perceived, as suggested by Social Cognitive Career Theory [18]. Yet, we lack an in-depth understanding of how URiM students experience this critical timeframe [7]. To address this gap, we undertook a qualitative exploration of URiM students at the end of their first year of medical school to characterize experiences that students described as influencing how they understand and approach research careers. Such insights may be crucial for informing early interventions and educational strategies to support URiM students' development as future physician-scientists, ultimately helping to build a more innovative and impactful biomedical research workforce.

## Methods

### Study design and sample

This qualitative descriptive study is a component of an ongoing 5-year, longitudinal, mixed-methods parent study (LEAP Study) that aims to identify trajectories and student- and systemic-level predictors of RCI among MD students [19]. The qualitative component explores students' lived experiences described as influencing RCI at key time points across training. This manuscript presents analyses from interviews conducted with URiM students at the end of the first year of medical school, offering insights into experiences influencing the early promotion or loss of RCI.

LEAP parent study participants were first-year MD or MD/master's students enrolled at any Liaison Committee on Medical Education (LCME) accredited U.S. or U.S territory medical school during academic year 2024−25 (URiM n=883; non-URiM n=253) [19]. MD-PhD students were excluded [19]. We identified a random sample (n=90) of URiM participants and employed stratified sampling by self-reported race/ethnicity, sex, gender, level of RCI, as well as by verified institution and geographic region, to ensure representativeness. RCI was defined using a modified two-part item from the AAMC Graduation Questionnaire: participants were first asked which career activities they planned to pursue (e.g., patient care, research, teaching, administration), and those selecting "research" were then asked how extensively they expected to be involved ("limited," "significant," or "exclusive") after completing their training, indicating their level of RCI. This measure at graduation has been shown to predict later NIH mentored-K and R01 award receipt [20]. We used the NIH and Association of American Medical Colleges (AAMC) definition of URiM as racial and ethnic groups whose representation in medicine lags far behind their share of the general population, including participants identifying as Black or African American, Hispanic/Latinx, AIAN, NHPI or multiracial including one of these races and ethnicities [7]. Students who self-identified as AIAN or HNPI were considered to be Indigenous-identifying.

### Recruitment

We sent recruitment invitations in two waves from our pre-identified random sample of 90 URiM participants: an initial wave of 60 on April 21, 2024 (spring of the first year of medical school) which yielded enrollment of 28 participants, followed by a second wave to the remaining 30 which yielded enrollment of four more participants (total enrolled=32). One respondent ultimately declined to be interviewed due to scheduling conflicts (total interviewed=31); reasons for the non-response among others were not assessed. Recruitment ended on May 9, 2024. Interested participants were sent the consent form (detailing the study aims, voluntary nature of participation, confidentiality protocols etc.) for review in advance of their interview. During the interview, the interviewer read and reviewed the consent language with the participant, invited questions, and asked if they agreed to proceed. Participants were informed that they could choose not to participate (with no professional or other consequences) and choose to end the interview at any time. We documented the date informed verbal consent was obtained in REDCap. The Yale University Institutional Review Board approved all study procedures (ID: 2000037712).

### Data collection

We developed and pilot tested an in-depth interview guide consisting of questions to elicit information about students' professional identity, experiences before and during medical school that influence career intentions, including RCI, and

the impact of race and/or ethnicity on experiences in medical school. We used probes to clarify and encourage detailed descriptions of participant experiences (see S1 Table). Interviews were conducted in May 2024.

Five members of our multidisciplinary research team (MOC, AH, MN, DB and SC)—four women and one man, comprising faculty and trainees in academic medicine, some identifying as URiM—conducted one-on-one interviews via Zoom in private rooms, with no others present. Interviewers had no personal or evaluative relationships with participants; some may have had brief prior contact with MOC through parent-study communications. All team members had prior experience leading or contributing to funded projects and peer-reviewed publications employing qualitative methods. Participants were informed that the study aimed to explore URiM students' experiences and factors shaping their RCI. Interviewers introduced themselves as members of the LEAP research team with experience in qualitative research and no evaluative relationship to participants. Interviews lasted approximately 45 minutes, and no repeat interviews were conducted. Participants received a $50 virtual Amazon gift card upon completion. We audio-recorded and transcribed interviews using Temi automated transcription software (www.temi.com) and reviewed all audio files and transcripts to ensure accuracy and confirm data quality. Transcripts were not returned to participants for comment or correction, because post-interview member checking was not integral to the study design. Field notes were made during and immediately after each interview to capture contextual observations and initial analytic impressions. Interviewers engaged in brief reflexive debriefings after each session to consider how shared identities and professional roles could have influenced rapport and questioning.

## Analysis

This study followed a qualitative descriptive orientation grounded in a constructivist paradigm, employing the constant comparative method to inductively identify patterns and themes across interviews [21,22]. An eight-member, racially and ethnically diverse, multidisciplinary research team used a systematic, iterative process to develop a coding structure. Working independently, each team member reviewed four transcripts and generated preliminary codes inductively. The team met to compare and refine these initial codes, developing an initial coding framework that was then applied to the remaining transcripts by four members working in pairs. Coding pairs met regularly to reconcile discrepancies and refine code definitions as new ideas emerged. Inductive coding produced a final structure comprising 17 codes and 39 subcodes; that was applied to all transcripts (see S2 Table). We identified patterns within and across interviews; some patterns coalesced as prominent themes supported by robust data and central to our primary research question. When a subset of data reflected a distinct but related dimension of a theme, illustrating meaningful variation or elaborating a specific mechanism, we designated these as subthemes. To ensure analytic adequacy, we assessed information power rather than aiming for data "saturation", consistent with Varpio et al.'s call to replace saturation with theoretically congruent concepts [23]. We judged our sample to have sufficient information power because (1) the study aim was focused; (2) the sample was specific to URiM students in their first year of medical school; (3) interviews elicited rich, detailed narratives; and (4) analysis was iterative and comparative, allowing exploration of both common and divergent experiences. These features collectively provided theoretical sufficiency—that is, the data were "good enough" to support robust, transferable insights, in contrast to the "exhaustive" notion implied by saturation [23].

We used ATLAS.ti Web (version 24) [24] for data management, coding, and analysis. To enhance trustworthiness and rigor, we maintained a detailed audit trail documenting analytic decisions, code refinements, and theme development, reviewed all transcripts against audio recordings for accuracy, and engaged in reflexive team discussions to ensure that interpretations reflected participants' voices rather than researcher presuppositions, acknowledging how our own positionalities as faculty and trainees in academic medicine might shape interpretation. Identifying details in participant quotations were minimally modified, when necessary, to protect confidentiality without altering meaning.

## Participant and public involvement

URiM medical students were involved at multiple stages of this study. The project was conceived by academic medicine faculty engaged in mentoring URiM medical students and informed by their observations of student engagement in research. URiM students

contributed to the refinement of research questions and served as the primary study participants. URiM students also participated in recruitment, conducted interviews, contributed to coding and analysis, and collaborated in authoring and revising the manuscript. The interview guide was piloted with a small group of URiM students to assess clarity and burden. All participants were financially compensated and invited to share feedback at the conclusion of their interviews, though they did not provide feedback on the study findings.

### Inclusivity in global research

Additional information regarding the ethical, cultural, and scientific considerations specific to inclusivity in global research is included in the Supporting Information (S1 Checklist).

## Results

Participant characteristics are shown in Table 1. We identified four major themes and five subthemes (Table 2) capturing how underrepresented students in their first year of medical school described experiences that influenced their interest in pursuing research careers. Themes included: (1) structured premedical research exposure was described as pivotal to developing early research engagement and interest in research careers; (2) research orientations reflected a commitment to using research as a vehicle for social justice and community impact; (3) high-quality research mentorship was characterized by authentic relational investment, skill development, and the distinct value of racial and ethnic identity-concordant role models; and (4) the research arms race for residency placement was described as amplifying systemic inequities that constrained students' research engagement.

### Theme 1: Structured premedical research exposure was described as pivotal to developing early research engagement and interest in research careers

**Subtheme:** Dedicated research offices or programs that support underrepresented students and/or those unfamiliar with research facilitated engagement in experiences students regarded as essential for becoming acclimated with research. Students described these opportunities as important entry points, providing both exposure and guidance. Such programs

**Table 1. Participant characteristics.**

| Characteristic | N (%) or Mean (SD) |
| --- | --- |
| **Medical school year 1** | 31 (100%) |
| **Age (years)** | 24.8 (2.6) |
| **Sex assigned at birth** | |
| Female | 20 (64.5%) |
| Male | 11 (35.5%) |
| **Race and ethnicity** | |
| Black | 12 (38.7%) |
| Hispanic | 10 (32.3%) |
| Multiracial | 9 (29.0%) |
| **Institution ownership** | |
| Private | 15 (48.4%) |
| Public | 16 (51.6%) |
| **Geographic location of institution** | |
| Midwestern U.S. | 6 (19.4%) |
| Northeastern U.S. | 7 (22.5%) |
| Southern U.S. | 6 (19.4%) |
| Western U.S. | 12 (38.7%) |

Indigenous-identifying students (AIAN and NHPI; see Methods) are represented within the Multiracial category (n=4, 12.9%).

**Table 2. Themes and subthemes describing experiences shaping research career intention among first-year URiM medical students.**

| |
|---|
| **Theme 1: Structured premedical research exposure was described as pivotal to developing early research engagement and interest in research careers.** |
| *Subtheme:* Dedicated research offices or programs that support underrepresented students and/or those unfamiliar with research facilitated engagement in experiences students regarded as essential for becoming acclimated with research. |
| *Subtheme:* While students with premedical research exposure reported early interest in research, barriers such as limited mentor engagement, lack of acknowledgement, and feeling undervalued as collaborators dampened their motivation to pursue it. |
| **Theme 2: Research orientations reflected a commitment to using research as a vehicle for social justice and community impact.** |
| **Theme 3: High-quality research mentorship was characterized by authentic relational investment, skill development, and the distinct value of racial and ethnic identity-concordant role models.** |
| *Subtheme*: Mentor-mentee racial and ethnic identity concordance was described as supporting feelings of belonging, encouragement, and a trusted space to share experiences, expanding students' vision of themselves as physician scientists. |
| **Theme 4: The research arms race for residency placement was described as amplifying systemic inequities that constrained students' research engagement.** |
| *Subtheme*: Research productivity and prestige metrics for residency conflicted with students' values and amplified structural inequities such as the minority tax and financial constraints. |
| *Subtheme:* Inequities in social capital constrained students' navigation of research pathways and contributed to feelings of marginality within medicine. |

both supported involvement in and demystified research. One student described a research program for Latino students which "helped paint the big picture", from building effective mentoring relationships to understanding the variety of research pathways.

> *"They gave us workshops on how to write an abstract or what a good mentor or PI-RA type relationship should look like or what the progress of a project should look like, the different roles of the people in your lab... I think that was very helpful for me to frame what research looks like because there's many different types of research... there's your basic science, your clinical, there's translational... They helped paint the big picture for us, especially for those who didn't know what research looked like to begin with."* (ID 1891: Hispanic female)

Another student described these programs as "opening doors" and giving them the first research opportunity that enabled them to pursue additional opportunities prior to matriculating into medical school. This student later reflected that their challenges finding and participating in research may not be the experience of peers who are not from underrepresented backgrounds.

> *"I think there's a lot of barriers for undergrads to be involved in research... I can't say that's the same case [for] students who... aren't technically underrepresented in medicine who have connections in the system already."* (ID 226: Asian/ NHPI male)

**Subtheme:** While students with premedical research exposure reported early interest in research, barriers such as limited mentor engagement, lack of acknowledgement, and feeling undervalued as collaborators dampened their motivation to pursue it.

Navigating research experiences with minimal guidance is a common challenge that students may encounter at some point in their training. One student expressed the degree of autonomy as challenging, particularly coming in without direction:

*"...Overall [the research lab] was not like the best experience because I was new to research and [it] didn't give a lot of direction... it actually made me kind of not want to do wet lab research as much... I think the autonomy was a challenge just because I didn't have as much direction coming into it."* (ID 1237: AIAN/Asian male)

Another described the experience of not receiving acknowledgement for their research efforts and being denied authorship, which made them feel less inclined to pursue research.

*"There was a situation towards the end of my experience where I thought I was going to be in a publication... unfortunately, the PI said I wasn't going to, even though I dedicated a year and a half to the project. So that really made me pretty disappointed. And that was out of my control... I think that just made me not really want to do research..."* (ID 1365: Hispanic female)

Some URiM students believed they had fewer options than peers from privileged backgrounds. The lack of alternative premedical research opportunities fostered a sense of resignation to remain in less beneficial ones, as shared by this participant:

*"That's a year [of research] that…frankly kind of went to waste... I also feel like that's a more common experience for students with similar background as myself because we don't really have other avenues. And sometimes we will just try to stick with what isn't really working out and benefiting us for the sake of fear of losing something that could help us further along our career."* (ID 226: Asian/NHPI male)

### Theme 2: Research orientations reflected a commitment to using research as a vehicle for social justice and community impact

A prominent sentiment was the intention to pursue research not solely as an academic endeavor, but as a meaningful tool to drive broader societal change. Students articulated diverse areas of interest, spanning basic science and translational research, community-based public health, policy advocacy, and global health. Their research motivations were often grounded in their identities and lived experiences, particularly a desire to address health inequities affecting marginalized populations.

For instance, one student discussed entering medical school with an interest in educational research, but shifting toward health services and health systems research, a transition informed by their upbringing in marginalized communities:

*"I came in with the idea of that I wanted to learn about educational research… but actually being in medicine, I was drawn to other aspects of research…that might have to do more with health services and health systems research…A lot of my interest in medicine is informed by growing up in marginalized communities…I feel more empathetic to the plight of other marginalized groups…growing up that way also gave me better insight into what other groups…might be going through. Not to say that I understand it. That's the point of the research, right? To really use that as a mechanism to have a better understanding of it. But the goal in the end is to bolster everybody up."* (ID 130: Hispanic male)

Students who had opportunities to conduct research with populations they identified with or sought to serve, described these experiences as personally meaningful, reinforcing their commitment to pursue research as part of their future physician identity. One student described that studying mental health within the Latinx community broadened their understanding of what research could be, showing that meaningful inquiry "doesn't have to be wet labs" but can center on community needs, and that this realization "really push[ed] [them] to prioritize that as a physician." (ID 292: Hispanic female) Others echoed this sense of purpose, describing inspiration drawn from community-based participatory research:

*"We work directly with the community and look into things that they wanted us to look into. At the end of the day, that's why we do research, to help patients…...how can we as an institution help better the relationship with people in the community? How do we better serve our community?... And that really inspired me."* (ID 1916: Black female)

A consistent narrative emerged about research serving as a foundation for advocacy, policy change, and broader societal transformation, particularly in low-resourced contexts:

*"… I feel as though if we want to create change, we have to do it through laws and policies to create a long-lasting impact… I was able to participate in clinical research…And it was then where I saw firsthand how research can improve standard of care for vulnerable populations as well as their families…Sometimes you think clinical care is the only way that we can impact patients' lives directly…"* (ID 6: Black male)

**Theme 3: High-quality research mentorship was characterized by authentic relational investment, skill development, and the distinct value of racial and ethnic identity-concordant role models**

Positive relationships with mentors were described as supporting students' interest in research. Participants valued research mentors who view mentorship as a critical part of their role in academia, and who demonstrate a genuine investment in student professional development. One student remarked, "This person is not just motivated to have med student labor but is really motivated in sort of supporting medical student trajectories." (ID 145: Hispanic male) Many students perceived the mentor-mentee relationship as applying to life outside their research and medical training. Students commonly expressed appreciation for mentors taking an interest in their personal lives and overall well-being. They welcomed getting to know mentors personally; for example, students commented, "… she met my parents…[has] gone above and beyond in that way. Because she knew I was out here on my own," (ID 1604: Black/Hispanic female) and:

*"The mentor that I have there, she's super cool…somehow we have like the same interests. She loves to sew. And she actually said that, this block she can teach me to sew on the side"* (ID 1365: Hispanic female)

Students described thriving when mentors provided a nurturing environment for research skill development. This meant meeting students "where [they are]," whether they arrive without previous research experience or with highly developed research skills. Students valued the freedom to "know nothing," (ID 1775: Hispanic/White male) make mistakes, and ask questions.

*"I think having that relationship…where your mentee can ask whatever they want and not feel embarrassed or just scared to ask you…that's important in a mentor."* (ID 2046: Black female)

Mentors provided exposure to opportunities across the research landscape. As one student noted, "it's just the networking"—several students described how mentors connected them to additional research opportunities and potential mentors, including near-peer mentors.

*[Faculty mentors facilitate access to] "upperclassmen who bring me on to their project are very willing to teach me, they're willing to show me the ropes…[an upperclassman] was just showing, giving me tips and... advice on writing a manuscript…"* (ID 226: Asian/NHPI male)

Students described how exposure to research mentors broadened their understanding of academic medicine and allowed them to imagine a career involving research. When asked what types of experiences during medical school might influence their interest in research, several participants commented on the importance of physician-scientist role models:

*"They are the blueprint in a sense... Because you can see, do I like what they're doing? Do they look happy? Are they enjoying it? How busy are they... And if you can see yourself in what they're doing, that's a big influential factor."* (ID 1979: Hispanic female)

**Subtheme:** Mentor-mentee racial and ethnic identity concordance was described as supporting feelings of belonging, encouragement, and a trusted space to share experiences, expanding students' vision of themselves as physician scientists.

Nearly all students whose research mentor shared their racial/ethnic identity described this concordance as providing role modeling that was transformative, particularly in terms of their perceptions of belonging in research.

*"It was crazy for me to even see myself or just Black men in general within medicine. But to take it a step further and be a Black man in medicine and doing research, not only have I never seen it, I just didn't think it was possible. And so that has been one barrier that has been kind of just completely squashed."* (ID 18: Black male)

Many students who described experiences of lacking a mentor with shared racial/ethnic identity reported feelings of otherness, academic spaces. Not seeing individuals who looked like them raised questions regarding the training and skills they would require to be successful in research.

*"it's all about seeing people like me in [research] and then seeing how it's possible and then also knowing how to get into it… if I say I want to be a researcher…it's a bit hard to visualize... how do you get funded… am I applying for a specific research topic or to satisfy a specific research need... how much support is offered to incoming PIs? What does the world of research look like?"* (ID 90: Black female)

Students described racial and ethnic identity-concordant mentors as uniquely understanding and affirming, fostering self-confidence, and creating trusted spaces to share perspectives shaped by their background and lived experiences. One student explained that "representation matters because having a mentor who understands where you're coming from and your lived experience firsthand really does make a difference," adding that they "feel more encouragement from them." (ID 226: Asian/NHPI male) Another student reflected that having a Black, male mentor helped dismantle internalized doubts, showing that "a lot of the barriers that I thought were barriers aren't actually barriers," and that they could "be in a lot of spaces and do a lot of things, like research." (ID 18: Black male) Another student described the deep trust and openness that emerged in her relationship with a mentor of shared background:

*"We speak the same languages and we have a similar background. And so I really trust her with things like this because I know that she's experienced similar in her training. Being able to talk with her about that and voice my observations and sometimes my frustrations with someone who I definitely know has experienced the same thing or similar and has a point of view on how she dealt with that has been really helpful."* (ID 1950: Hispanic female)

Students also identified racial and ethnic identity concordance as central to envisioning what their life as a physician scientist might look like. When asked whether there was anything about medical school training that would make them more likely to consider a career as a physician scientist, one student remarked:

*"If I had a Black female mentor that was a physician scientist and I established a close relationship, 100%... I don't have a lot of Black female mentors… if I kind of had more representation to what that looks like for someone who looks like me, I think I would consider it more so than I am now."* (ID 2046: Black female)

However, interest in seeking out racially or ethnically concordant mentors was not universal. One student suggested that having shared identity with a mentor wasn't a concern, that they felt faculty efforts to connect them with mentors of a similar background were unnecessary, commenting:

*"She's really nice, but I feel like she was more guiding me towards check[ing] out these… Black professors or... just Black people in power or something, probably to help me relate. But... I don't always look at the color when I want a mentor. It's more so, oh, I really admire this about you. I want to be surrounded by you. And so I was like, okay, whatever. I'm not gonna seek out those connections but thank you."* (ID 90: Black female)

### Theme 4: The research arms race for residency placement was described as amplifying systemic inequities that constrained students' research engagement

Over half of all students described a research arms race—pressure to publish prolifically in high impact journals to remain competitive for residency. Students emphasized that these productivity and prestige metrics amplified existing inequities, including the minority tax, financial constraints, and unequal social capital, shaping what research felt feasible and how they related to it.

**Subtheme:** Research productivity and prestige metrics for residency conflicted with students' values and amplified structural inequities such as the minority tax and financial constraints.

Students described how the expectation to publish prolifically and in high-impact journals for residency transformed research from an avenue of inquiry and purpose into an instrumental task akin to a "checkbox" (ID 145: Hispanic male). Several felt that these expectations were misaligned with their values of service and community impact, and that they deepened existing structural inequities rather than rewarding authentic engagement.

Students explained how the pressure to demonstrate research productivity collided with an already unequal landscape of expectations for underrepresented trainees, who "when they enter spaces or institutions where it's an anomaly or a privilege for them to be there", must balance both the "responsibilities expected of them as a student, and "the extra work of giving back to the community that got them there in the first place." (ID 837: Asian/NHPI male) Several identified the research arms race as intensifying this minority tax, especially because the labor of community-oriented work that, while deeply meaningful, may be undervalued in academic promotion systems that reward volume and prestige.

*"I think it [community health research project with Pacific Islander groups] also falls a little bit into like the minority tax kind of concept of things where I am interested in doing all of that, but it may not yield like the highest publications, or it may not yield the most…high impact journals… will it also negatively impact my future of being able to match elsewhere…Or matching some more competitive specialties that I may be interested in? So, by pursuing that, am I closing the door for my future self?"* (ID 837: Asian/NHPI male)

Others described how inequities in privilege constrained the very ability to navigate the research environments. One student observed that peers with inherited confidence and security could set limits that underrepresented students could not:

*"Some students, they have the privilege and attitude of 'Oh, I can take on these projects. But also I see myself as an asset in that if I don't wanna do this particular part, I could just tell my PI, oh, I don't wanna do this', you know? Whereas us underrepresented [students], we wanna do everything and anything that we can. And then sometimes we over commit and then we get all stressed…"* (ID 1946: Hispanic female)

Students also pointed to the financial costs of these expectations, particularly advice to take an unpaid research year to remain competitive for residency. One student called this advice "bordering on injustice", noting:

*"It is just the economics that minorities, as a whole live with day to day." The same student continued "Many of us apply to medical school understanding that it's a four-year program. And I don't think it's acceptable that they're thrusting the burden of paying a fifth year because Step [licensing exam] went pass-fail [resulting in the need for research engagement to distinguish themselves]. The changes shouldn't be shouldered by us financially or with our time."* (ID 130: Hispanic male)

These pressures collectively left students questioning the moral integrity of research as it is currently rewarded. Some expressed dissatisfaction with "easy paper(s)" pursued to meet perceived residency metrics, where "[they're] not adding anything", stating that "it is the actual contribution that makes it worthwhile—not just doing papers for the sake of it and having the numbers to be competitive." (ID 1373: Black male) Others described growing disillusionment altogether, explaining that volume-based expectations stripped their work of meaning and authenticity:

*"When you have to have 30 publications or you're only competitive if you've worked with this type of person and you've done this type of work. I just think that is ridiculous. I found that it's made me really indifferent to what I'm doing as long as I'm doing it…I don't end up doing things that I'm genuinely interested and passionate about because I'm just worried about volume and about what I'm gonna look like on paper. And I don't like that…I don't really identify with a physician scientist role."* (ID 1021: Black female)

For some, these reflections extended to questioning whether the broader culture of academic medicine could ever align with their moral orientation toward social justice:

*"[I'm] just not feeling like it's value aligned. Like, oh, I'm gonna do all of this stuff so that I can get this higher position. That kind of culture of academic medicine doesn't sit well with me…I love mentorship and I wanna work with people who are in training. But if there are so many restrictions, I don't know…The physician who runs [the Diversity, Equity, and Inclusion office at the students' institution], she's one of my biggest mentors here. And I love what she does, but again, I think she's limited in the way sometimes of how she so shows us support because of the position she holds…"* (ID 292: Hispanic female)

**Subtheme**: Inequities in social capital constrained students' navigation of research pathways and contributed to feelings of marginality within medicine.

Students described social capital as familiarity with medicine, access to people who could open doors, and insider knowledge of how success in medicine is built—advantages they noted were enabled by family wealth and inherited privilege. Those without college-educated or employed parents emphasized lacking the informal guidance, early exposure, and introductions that helped peers from more privileged backgrounds obtain research mentors and authorships. Many spoke of having to "figure everything out on [their] own" (ID 1365: Hispanic female) and struggling to seek support, with one student linking this to "not know[ing] what support might look like" (ID 2097: Black/White male).

*"[I'm] kind of starting to realize that most of my classmates are in a much different tax bracket than I am and have both parents that are doctors. And so they've got someone to ask every single question to and they've been exposed to this field for a lot longer than I have. And that they have these like multi-year long relationships with mentors…who are physicians that have been advocating for them since before undergrad. So just seeing the really stark difference in our paths to medicine and seeing how much of a leg up they had. It has been a little challenging for me to feel like I'm on par with that…When we're having to learn things or we're going through difficult times, it's a little harder for me to go and seek out that support. And for them, they've had it already."* (ID 1950: Hispanic female)

Students explained that lacking this social capital meant "learning through trial and error" and "put[ting] more effort" (ID 226: Asian/NHPI male)—being "on top of [their] emails" (ID 1950: Hispanic female), taking on leadership roles to be able

to meet mentors, "cold emailing physicians" (ID 1679: Black female) to enquire about research opportunities, "go[ing] out of [their] way" and "work[ing] twice as hard" (ID 28: Hispanic female) to make connections their peers inherited. As one student put it, "you're trying to do everything that everybody else is doing but also make up for your lack of social capital." (ID 1722: Black male) Several wondered how they could possibly compete with more privileged peers in the research arms race:

> "Even though my parents were physicians in [South American country], they really just didn't understand how the education system works here… I went through the whole process blind… I have one classmate who has a sibling in the specialty they wanna match into… they already have 20 publications in the first four months… because the residents that are friends with that sibling are just putting them on the papers. And it's like—how do you compete with that?" (ID 1775: Hispanic/White Male)

Students also reflected on how these unequal starting points and not having "anyone to relate to" (ID 1664: Black female) on their professional journey influenced their emotional experience of medical school. Some described feeling "a sense of shame," or "intimidate[ed]" (ID 1021: Black female) when comparing themselves to peers with generational access and realizing how far behind they began. For some, this inequity in social capital deepened feelings of imposter syndrome:

> "I think most people have imposter syndrome, but sometimes I really feel it in medical school because… a lot of people talk about how their parents are physicians or they have people in their family who's a physician…I kind of feel small in a space where we are essentially… in the same race. But it's very clear to me that we started at different points in life… Medicine is honestly a very elitist field…It's really hard to break into…" (ID 226: Asian/NHPI male)

## Discussion

### Overview of key findings

This study offers a detailed account of how RCI begins to take shape among URiM students during their first year of medical school. Our findings highlight four interconnected domains of experience that participants described as salient in how they made sense of early research engagement and research career possibilities: (1) exposure to structured research opportunities before medical school, (2) the presence and quality of mentorship, particularly identity-concordant mentors, (3) perceptions of research as a vehicle for social change, and (4) structural inequities such as the minority tax, and unequal financial and social capital that are intensified by the research arms race for residency.

These findings extend the literature by capturing how these experiences unfold at a formative period seldom examined in depth, particularly via qualitative inquiry. While prior work has examined URiM-specific barriers to academic medicine careers more broadly [11,25], few studies have focused on the early medical school period or investigated how RCI specifically begins to emerge and evolve. Our data, drawn from interviews at the end of students' first year, illuminate how students' experiences—filtered by their values, identities, and institutional contexts—influence how they make sense of academic research and weigh whether a research career feels viable, purposeful, or aligned with their long-term goals. Notably, even at this early stage of their training, participants reflected on their research experiences using interpretive lenses of values alignment, structural inequity, and residency competitiveness.

### Early research exposure: Gateway and double-edged sword

Our results are consistent with prior literature in which students described pre-medical research exposure as contributing to their perceptions of technical skills, confidence, and self-efficacy [26]. Structured programs and research offices

provided critical "academic social capital" — teaching students how to find mentors, write abstracts, and navigate research processes. Prior research indicates that programs that provide underrepresented students with such early research training strengthen research skills [27] and may increase entry of URiM students into physician-scientist training programs [28]. Participants described structured research opportunities as especially important in supporting their confidence and navigation of research spaces, particularly for students who perceived themselves as lacking prior exposure or professional networks. At the same time, early research exposure may deter interest when experiences are marginalizing. The lack of diverse role models in pre-medical research environments can amplify feelings of not belonging [26]. Prior work supports that identity-concordant role models enhance URiM students' persistence in STEM, while their absence undermines belonging [29,30]. Our study suggests that the quality and inclusivity of early research experiences matter profoundly. Affirming mentored opportunities can level the playing field, while negative exposures risk reinforcing disparities.

### Altruistic motivation and diverse interests: Research as a tool for social change

Students described interests ranging from basic and translational science to research grounded in community-based participatory approaches. A striking finding was the altruistic motivation many URiM students described—viewing research as a mechanism to improve the health of vulnerable populations—which aligns with prior research [25,31,32]. URiM physicians are more likely to serve underserved populations [33,34] and several of our participants similarly gravitated toward research projects addressing health inequities among marginalized peoples. This motivation can enrich science by generating novel inquiry that substantially improves health outcomes in marginalized populations [35,36].

### The critical role of mentorship and representation

Consistent with literature on career development, mentorship emerged as a central interpretive theme in participants' accounts. This aligns with evidence that strong mentorship boosts confidence, productivity, and career satisfaction [37,38]. For URiM students who spoke of lacking informal networks, such relationships were described as transformative, consistent with prior literature [37]. Identity-concordant mentorship was described by participants as particularly meaningful in fostering a sense of belonging and possibility within research spaces, aligning with literature showing that representation supports persistence in STEM and academic careers [29,37]. Expanding the pool of URiM mentors is thus a structural imperative. That said, some evidence indicated appreciation of shared interests over shared identity, highlighting that cross-racial mentorship can also be highly effective when culturally responsive [37]. However, effective cross-cultural mentorship requires skill, and only ~26% of STEM doctoral programs offer formal mentor training [39]; the prevalence of such training among academic medicine faculty is unknown.

### The research arms race: External pressures influencing engagement

Students frequently described research as less a calling than a credential for career advancement. There has been increased emphasis on research productivity as residency program directors become less reliant on metrics such as standardized test scores [40–42]. By the early 2020s, successful applicants in fields like dermatology and plastic surgery reported more than 20 research products on average [43]. Our findings suggest that pressures associated with these benchmarks (which our study participants were acutely aware of) are already influencing how students approach and evaluate research as early as the first year of medical school. This trend raises two concerns. First, it may prioritize quantity over quality, contributing to concerns the research arms race dilutes research integrity [44]. Second, it may amplify inequities for URiM students. Despite participating in a similar number of research experiences, URiM students have lower publication rates than their non-URiM peers [45], a disparity influenced by reduced access to effective mentorship, sponsorship, professional networks, and other forms of academic social capital [11,14]. Further, many non-URiM peers rely on financial means to take an unpaid research year—resources not equally available to many URiM students [45]. Moreover, some URiM students experienced a student-level minority tax: a sense that essential research pursuits are piled on top of

commitments that non-URiM students may be less likely to engage in, namely work to "give back" to the community and paid employment. National data echo this tension: URiM students spend significantly more time than peers on advocacy and diversity, equity, and inclusion work, which can detract from other pursuits [46]. These dynamics risk discouraging students who could enrich academic medicine with diverse perspectives. Systemic changes are needed: reevaluating residency selection to prioritize quality and authentic engagement, expanding paid research opportunities, and providing mentoring and sponsorship for students without inherited social capital.

### A sense of value misalignment and research disengagement

While our findings span distinct experiential domains, a consistent thread was students' references to perceived value incongruence when reflecting on their research experiences, which influenced how they articulated enthusiasm, ambivalence, or hesitation towards future research engagement. Many questioned whether prevailing research culture and academic promotion systems aligned with their values of service to marginalized populations and advancing science. Some described this perceived misalignment as dampening intrinsic motivation; and others described it influencing their perception of research as a strategic obligation rather than a meaningful pursuit. In this context, value-concordant mentorship that extended beyond academic guidance—affirming students' identities, backgrounds, and aspirations—was especially influential. Notably, students described these value tensions as emerging during the first year of medical school, highlighting that questions about alignment with academic research culture can arise early in medical training. These findings suggest that fostering research engagement among URiM students may require not only expanding access and skills but also aligning research environments with students' core values and motivations for pursuing scholarly work.

### Limitations

First, our qualitative sample of 31 MD students, while drawn from a national cohort, may not capture the full spectrum of URiM student experiences. Participation was voluntary, potentially attracting students with stronger opinions about research (either positive or negative). However, achieving theoretical sufficiency within a relatively homogenous sample provides confidence in the robustness of our findings [23]. Second, our self-reported data may be influenced by recall bias or social desirability. Even with established approaches to promote candor and a diverse interviewer team, interviews may not have fully captured all sensitive experiences, including experiences of bias. Conducting interviews virtually may have further constrained rapport and depth of disclosure. Third, while our study identifies experiences influencing RCI among first year students, it may not capture factors shaping the evolution of RCI over the course of medical training. Our ongoing longitudinal study will follow these same students to examine how their perspectives develop throughout medical school. Fourth, consistent with a qualitative descriptive approach, the concepts referenced such as social cognitive career theory, identity concordance, minority tax, social capital, and values alignment, are used to interpret participants' accounts rather than to imply causal relationships or predict research career trajectories. Lastly, we have aimed to mitigate biases through team-based coding and reflexive analysis, but we acknowledge our own interpretive lens as researchers committed to diversity in medical education.

### Implications

This study offers actionable guidance for medical educators, academic leaders, and policymakers working to diversify the physician-scientist workforce. First, our findings highlight URiM medical students' perceptions of the value of structured, affirming pre-medical research exposure. Medical schools and funders should continue investing in pathway programs that proactively engage URiM students before and during medical school. These programs should emphasize not just placing students in labs but ensuring high-quality experiences – by training PIs to integrate and credit novice researchers. Financial support through stipends and scholarships are essential to ensure access for students from lower-income backgrounds.

Second, mentorship emerged as a cornerstone for URiM physician-scientist development. Students described thriving under mentors who offered not only research guidance but also authentic, affirming relationships that acknowledged their backgrounds, aspirations, and full identities. Institutions should invest in mentorship structures that promote these qualities. This could include pairing students with near-peer or faculty mentors who have demonstrated success mentoring URiM trainees and offering students the option to be matched with mentors of similar racial or ethnic backgrounds when desired. Given the limited pool of URiM faculty, multiple-mentor models (combining content-area expertise with identity-concordant support, for instance) may be especially beneficial. Faculty should be supported in developing inclusive mentoring practices that affirm students' lived experiences and motivations, not just their research productivity. In settings where local mentorship resources are limited, national professional societies could help facilitate cross-institutional mentorship, including virtual connections to identity-affirming mentors in students' fields of interest.

Third, institutions should consider re-evaluating how research expectations tied to residency competitiveness may marginalize URiM students. Participants described how pressure to produce high-volume outputs in prestigious journals amplified structural barriers—including financial strain, limited social capital, and the minority tax—and often conflicted with their values. Some students questioned whether equity-driven or community-engaged research, despite its personal significance, would carry equal academic currency in residency selection, raising concerns about trade-offs between meaningful engagement and competitiveness. To address this, academic leaders should broaden definitions of scholarly productivity to explicitly recognize and reward diverse forms of research (particularly in specialties that have historically prioritized bench science); provide protected time for substantive inquiry without extending training; and adopt residency review practices that assess the substance, contribution, and context of students' work. Importantly, cultivating research environments that affirm students' values and motivations for scholarly engagement—rather than reducing research to a checkbox for competitiveness—may better support meaningful scholarly engagement.

## Conclusion

This qualitative study offers an in-depth examination of experiences first-year URiM medical students described as informing their engagement with research and how they weighed the possibility of physician scientist careers. Students described how structured premedical research exposure, community-oriented motivations, invested mentorship (especially from identity-concordant mentors) and structural barriers intensified by residency-driven publication pressures influenced how they engaged with research and evaluated research careers. Notably, even at this early stage of medical training, students described tensions between their personal values and prevailing academic research culture, framing research less as a meaningful pursuit and more as a strategic obligation.

These findings underscore the need for proactive institutional change: expanding access to early research opportunities; strengthening mentorship through relational, culturally responsive, and network-based models; and transforming academic promotion systems to affirm students' values and motivations for scholarly inquiry. Future research should examine how dynamics of value alignment and structural opportunity may evolve over time and influence the trajectory of research engagement and RCI across medical training. Advancing a more diverse and enduring physician-scientist workforce may require aligning research environments with the values, aspirations, and lived experiences of those historically excluded from academic medicine.

## Supporting information

**S1 Table. Interview guide.**
(DOCX)

**S2 Table. Code structure.**
(DOCX)

**S3 Table. Additional quotes supporting each theme and subtheme.**
(DOCX)

**S1 Checklist. Inclusivity in global research.**
(DOCX)

## Author contributions

**Conceptualization:** Alexandra M. Hajduk, Sarwat I. Chaudhry, Dowin Boatright.

**Data curation:** Shruthi Venkataraman, Meghan O'Connell, Allison Aviles.

**Formal analysis:** Shruthi Venkataraman, Meghan O'Connell, Adeola Ayedun, Allison Aviles, Alexandra M. Hajduk, Mytien Nguyen, Sarwat I. Chaudhry, Dowin Boatright.

**Funding acquisition:** David Henderson, Judee Richardson, Rachel K. Wolfson, Sarwat I. Chaudhry, Dowin Boatright.

**Investigation:** Shruthi Venkataraman, Meghan O'Connell, Adeola Ayedun, Alexandra M. Hajduk, Mytien Nguyen, Leslie A. Curry, John Paul Sánchez, Rachel K. Wolfson, Sarwat I. Chaudhry, Dowin Boatright.

**Methodology:** Shruthi Venkataraman, Meghan O'Connell, Adeola Ayedun, Alexandra M. Hajduk, Leslie A. Curry, Rachel K. Wolfson, Sarwat I. Chaudhry, Dowin Boatright.

**Project administration:** Shruthi Venkataraman, Meghan O'Connell, Allison Aviles, Sarwat I. Chaudhry, Dowin Boatright.

**Resources:** Gbenga Ogedegbe, Sarwat I. Chaudhry, Dowin Boatright.

**Software:** Sarwat I. Chaudhry, Dowin Boatright.

**Supervision:** Alexandra M. Hajduk, Leslie A. Curry, John Paul Sánchez, Rachel K. Wolfson, Sarwat I. Chaudhry, Dowin Boatright.

**Writing – original draft:** Shruthi Venkataraman, Meghan O'Connell, Adeola Ayedun.

**Writing – review & editing:** Shruthi Venkataraman, Meghan O'Connell, Adeola Ayedun, Allison Aviles, Alexandra M. Hajduk, Mytien Nguyen, Gbenga Ogedegbe, Laura Castillo-Page, David Henderson, Judee Richardson, Leslie A. Curry, John Paul Sánchez, Rachel K. Wolfson, Sarwat I. Chaudhry, Dowin Boatright.

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
