## [Decision Letter · Decision Letter 0]

24 Feb 2026

PONE-D-25-61426Experiences shaping research career intention among Black, Hispanic, and Indigenous first-year medical students in the United States: a qualitative studyPLOS One

Dear Dr. Venkataraman,

Thank you for submitting your manuscript to PLOS ONE. After careful consideration, we feel that it has merit but does not fully meet PLOS ONE’s publication criteria as it currently stands. Therefore, we invite you to submit a revised version of the manuscript that addresses the points raised during the review process.

We look forward to receiving your revised manuscript.

Kind regards,

Saima Aleem

Academic Editor

PLOS One

Journal Requirements:

“This study was funded by the National Institute of Health (NIH), project number R35GM153263. The NIH did not have any role in the study outside of funding. The authors listed are all independent from the NIH and had full access to all the data in the study and can take full responsibility for the integrity of the data and the accuracy of the data analysis.”

4. In the online submission form you indicate that your data is not available for proprietary reasons and have provided a contact point for accessing this data. Please note that your current contact point is a co-author on this manuscript. According to our Data Policy, the contact point must not be an author on the manuscript and must be an institutional contact, ideally not an individual. Please revise your data statement to a non-author institutional point of contact, such as a data access or ethics committee, and send this to us via return email. Please also include contact information for the third party organization, and please include the full citation of where the data can be found.

Reviewer's Responses to Questions

**Comments to the Author**

1. Is the manuscript technically sound, and do the data support the conclusions?

Reviewer #1: Yes

Reviewer #2: Yes

2. Has the statistical analysis been performed appropriately and rigorously? 

Reviewer #1: N/A

Reviewer #2: N/A

3. Have the authors made all data underlying the findings in their manuscript fully available?

Reviewer #1: No

Reviewer #2: Yes

4. Is the manuscript presented in an intelligible fashion and written in standard English?

Reviewer #1: Yes

Reviewer #2: Yes

5. Review Comments to the Author

Reviewer #1: This manuscript addresses an important and timely topic in medical education by exploring early experiences shaping research career intentions among students underrepresented in medicine. The qualitative design is appropriate, and the study is methodologically sound, with clear descriptions of sampling, data collection, and analytic processes. The findings are well supported by rich participant quotations, and the themes are coherent, analytically grounded, and meaningfully interpreted in relation to existing literature.

The manuscript is clearly written and well structured, and the conclusions are appropriately aligned with the data presented, without overstatement. The discussion thoughtfully situates the findings within broader debates on mentorship, equity, and the physician-scientist pipeline, offering relevant and actionable implications for educators and institutions.

One area that would benefit from further attention is data availability. While supplementary materials are provided, access to the full de-identified dataset is restricted and contingent upon request and completion of the parent longitudinal study. Greater alignment with PLOS ONE’s data-sharing policy, or a clearer justification for the stated restrictions, would strengthen transparency.

Overall, this is a strong and valuable contribution to the literature. With minor clarifications regarding data availability, the manuscript would be suitable for publication.

Reviewer #2: This manuscript presents a rigorous, thoughtfully conducted qualitative study examining early experiences shaping research career intentions among first-year students underrepresented in medicine. The study is clearly written, methodologically transparent, and robust empirical work with an appropriate analytical approach. The data are rich, and the themes are well supported by participant quotes. Overall, this is a strong paper that could benefit from minor revisions to further sharpen conceptual clarity and alignment between framing and findings.

First, the manuscript draws on several relevant concepts, including social cognitive career theory, identity concordance, minority tax, social capital, and values alignment. While all are appropriate, the Discussion occasionally assigns these concepts a stronger explanatory role than warranted by a qualitative descriptive design (mainly through words such as drives, determines, shapes etc.). I would recommend clarifying that these frameworks function as interpretive lenses rather than analytic or causal models, and slightly nuancing language that implies determination or directionality.

Second, the study's contribution and novelty could be made more explicit, for example, by briefly and clearly stating what this study adds beyond existing work, particularly the focus on the first year of medical school which is rarely examined, the early emergence of value misalignment with academic research culture, and the way residency competitiveness is already shaping research engagement at this stage of training.

Finally, the participant and public involvement section is a strength, but could be slightly streamlined to maintain a descriptive rather than evaluative tone. At the moment, it sounds more like advocacy, whereas the normative statements in this section would fit better in the Discussion.

These are relatively minor issues that do not detract from the overall quality of the work. With these revisions, the manuscript would make a valuable and methodologically sound contribution to the literature on medical education and research career development.

6. PLOS authors have the option to publish the peer review history of their article (what does this mean?). If published, this will include your full peer review and any attached files.

Reviewer #1: No

Reviewer #2: **Yes:**Sofiya Abedali

---

## [Author Response · Author response to Decision Letter 1]

13 Apr 2026

JOURNAL REQUIREMENTS

1.Please ensure that your manuscript meets PLOS ONE's style requirements, including those for file naming. The PLOS ONE style templates can be found at https://journals.plos.org/plosone/s/file?id=wjVg/PLOSOne_formatting_sample_main_body.pdf and

Response: We have edited the manuscript to ensure that it meets the PLOS ONE style requirements, including for file naming.

Response: We have now included a completed version of the PLOS ONE questionnaire on inclusivity in global research as Supporting Information and referenced it in the manuscript text, as requested.

Modified text (lines 212-214): “Additional information regarding the ethical, cultural, and scientific considerations specific to inclusivity in global research is included in the Supporting Information (S3 Checklist).”

3. Thank you for stating the following financial disclosure: “This study was funded by the National Institute of Health (NIH), project number R35GM153263. The NIH did not have any role in the study outside of funding. The authors listed are all independent from the NIH and had full access to all the data in the study and can take full responsibility for the integrity of the data and the accuracy of the data analysis.” Please state what role the funders took in the study. If the funders had no role, please state: "The funders had no role in study design, data collection and analysis, decision to publish, or preparation of the manuscript." If this statement is not correct you must amend it as needed. Please include this amended Role of Funder statement in your cover letter; we will change the online submission form on your behalf.

Response: The funders had no role in the study. We have now included the suggested statement verbatim in our cover letter.

4. In the online submission form you indicate that your data is not available for proprietary reasons and have provided a contact point for accessing this data. Please note that your current contact point is a co-author on this manuscript. According to our Data Policy, the contact point must not be an author on the manuscript and must be an institutional contact, ideally not an individual. Please revise your data statement to a non-author institutional point of contact, such as a data access or ethics committee, and send this to us via return email. Please also include contact information for the third-party organization, and please include the full citation of where the data can be found.

Response: We have amended our data statement to comply with both the PLOS ONE Data Policy and the terms of the verbal informed consent obtained from study participants.

Modified text (‘Data Availability’ statement): “Due to the highly sensitive nature of the data collected in this study, the data cannot be made publicly available in order to protect participant confidentiality and to comply with the terms of the verbal informed consent obtained from participants. At the time of consent, participants were informed that access to their information would be restricted to the research team and individuals responsible for research oversight, and that the data would not be shared with any parties outside the investigative team, including program leadership at their respective institutions. Questions regarding data access or participant protections may be directed to the Yale University Human Research Protection Program (HRPP@yale.edu).”

Response: We have included a full ethics statement with the full name of the IRB who approved our study and explained our informed verbal consent process in the ‘Methods’ section, as requested.

Associated text (lines 137-144): “Recruitment ended on May 9, 2024. Interested participants were sent the consent form (detailing the study aims, voluntary nature of participation, confidentiality protocols etc.) for review in advance of their interview. During the interview, the interviewer read and reviewed the consent language with the participant, asked if there were questions or answered questions and asked if they agreed to proceed with the interview. Participants were informed that they could choose not to participate (with no professional or other consequences) and choose to end the interview at any time. We documented that date informed verbal consent was obtained in REDCap. The Yale University Institutional Review Board approved all study procedures.”

Response: We have now included captions for the Supporting Information files at the end of the manuscript (see lines 841-844) and updated in-text citations (see lines 150, 180, 213-214, 216, 217) to match accordingly.

Response: The reviewers and editor did not make such recommendations.

Response: We have reviewed the reference list for completeness and accuracy. To our knowledge, we have not cited any retracted articles. However, we found a duplicated reference (# 14 and # 45 in the previous version) and have remediated the duplication in the revision.

REVIEWER’S COMMENTS

Reviewer #1:

1. One area that would benefit from further attention is data availability. While supplementary materials are provided, access to the full de-identified dataset is restricted and contingent upon request and completion of the parent longitudinal study. Greater alignment with PLOS ONE’s data-sharing policy, or a clearer justification for the stated restrictions, would strengthen transparency.

Response: We appreciate this important point. We have now amended the Data Sharing statement to reflect greater adherence with PLOS ONE’s data sharing policy and the terms of the verbal informed consent obtained from study participants.

Modified text (‘Data Availability’ statement): “Due to the highly sensitive nature of the data collected in this study, the data cannot be made publicly available in order to protect participant confidentiality and to comply with the terms of the verbal informed consent obtained from participants. At the time of consent, participants were informed that access to their information would be restricted to the research team and individuals responsible for research oversight, and that the data would not be shared with any parties outside the investigative team, including program leadership at their respective institutions. Questions regarding data access or participant protections may be directed to the Yale University Human Research Protection Program (HRPP@yale.edu).”

Reviewer #2:

1. First, the manuscript draws on several relevant concepts, including social cognitive career theory, identity concordance, minority tax, social capital, and values alignment. While all are appropriate, the Discussion occasionally assigns these concepts a stronger explanatory role than warranted by a qualitative descriptive design (mainly through words such as drives, determines, shapes etc.). I would recommend clarifying that these frameworks function as interpretive lenses rather than analytic or causal models, and slightly nuancing language that implies determination or directionality.

Response: We thank the reviewer for this insightful and constructive feedback. As suggested, we have nuanced and softened the causal-sounding language in the ‘Discussion’ as well as throughout the manuscript. We have also added a line in the ‘Limitations’ to explicitly clarify the epistemic limits of the work and findings.

Modified text (lines 522-524): “Our findings highlight four interconnected domains of experience that participants described as salient in how they made sense of early research engagement and research career possibilities…”

Modified text (lines 546-549): “Participants described structured research opportunities as especially important in supporting their confidence and navigation of research spaces, particularly for students who perceived themselves as lacking prior exposure or professional networks.”

Modified text (lines 571-574): “Identity-concordant mentorship was described by participants as particularly meaningful in fostering a sense of belonging and possibility within research spaces, aligning with literature showing that representation supports persistence in STEM and academic careers.[28,36]”

Modified text (lines 604; paragraph title): “A sense of value misalignment and research disengagement”

Modified text (lines 605-608): “While our findings span distinct experiential domains, a consistent thread was participants’ references to perceived value incongruence when reflecting on their early research experiences, which influenced how they articulated enthusiasm, ambivalence, or hesitation towards future research engagement.”

Modified text (lines 631-634): “Fourth, consistent with a qualitative descriptive approach, the concepts referenced such as social cognitive career theory, identity concordance, minority tax, social capital, and values alignment, are used to interpret participants’ accounts rather than to imply causal relationships or predict research career trajectories.”

Modified text (lines 674-676): “This qualitative study offers an in-depth examination of experiences first-year URiM medical students described as informing their engagement with research and how they weighed the possibility of physician scientist careers.”

Modified text (lines 678-679): “…and structural barriers intensified by residency-driven publication pressures influenced how they engaged with research and evaluated research careers.”

2. Second, the study's contribution and novelty could be made more explicit, for example, by briefly and clearly stating what this study adds beyond existing work, particularly the focus on the first year of medical school which is rarely examined, the early emergence of value misalignment with academic research culture, and the way residency competitiveness is already shaping research engagement at this stage of training.

Response: We appreciate this thoughtful comment and have added statements in the ‘Discussion’ and ‘Conclusion’ to explicitly highlight the more novel findings of this study.

Modified text (lines 529-538): “These findings extend the literature by capturing how these experiences unfold at a formative period seldom examined in depth, particularly via qualitative enquiry. While prior work has examined URiM-specific barriers to academic medicine careers more broadly,[11,24] few studies have focused on the early medical school period or investigated how RCI specifically begins to emerge and evolve…Notably, even at this early stage of their training, participants reflected on their research experiences using interpretive lenses of values alignment, structural inequity, and residency competitiveness.”

Modified text (lines 585-587): “Our findings suggest that pressures associated with these benchmarks (which our study participants were acutely aware of) are already influencing how students approach and evaluate research as early as the first year of medical school.”

Modified text (lines 614-616): “Notably, students described these value tensions as emerging during the first year of medical school, highlighting that questions about alignment with academic research culture can arise early in medical training.”

Modified text (lines 679-682): “Notably, even at this early stage of medical training, students described tensions between their personal values and prevailing academic research culture, framing research less as a meaningful pursuit and more as a strategic obligation.”

3. Finally, the participant and public involvement section is a strength, but could be slightly streamlined to maintain a descriptive rather than evaluative tone. At the moment, it sounds more like advocacy, whereas the normative statements in this section would fit better in the Discussion.

Response: We thank the reviewer for this helpful observation. In response, we revised the Participant and Public Involvement section to maintain a more neutral, descriptive tone focused on clearly documenting how URiM students were involved across stages of the study (e.g., refinement of research questions, recruitment, data collection, analysis, and manuscript preparation). We removed or avoided evaluative and normative language and limited this section to factual reporting of participant involvement. We believe these revisions better align the section with conventions for reporting participant and public involvement.

Modified text (lines 202-210): “URiM medical students were involved at multiple stages of this study. The project was conceived by academic medicine faculty engaged in mentoring URiM medical students and informed by their observations of student engagement in research. URiM students contributed to the refinement of research questions and served as the primary study participants. URiM students also participated in recruitment, conducted interviews, contributed to coding and analysis, and collaborated in authoring and revising the manuscript. The interview guide was piloted with a small group of URiM students to assess clarity and burden. All participants were financially compensated and invited to share feedback at the conclusion of their interviews, though they did not provide feedback on the study findings.”

---

## [Editor Report · Decision Letter 1]

27 Apr 2026

Experiences shaping research career intention among Black, Hispanic, and Indigenous first-year medical students in the United States: a qualitative study

PONE-D-25-61426R1

Dear Dr. Venkataraman,

We’re pleased to inform you that your manuscript has been judged scientifically suitable for publication and will be formally accepted for publication once it meets all outstanding technical requirements.

Kind regards,

Saima Aleem

Academic Editor

PLOS One
---

## [Editor Report · Acceptance letter]

PONE-D-25-61426R1

PLOS One

Dear Dr. Venkataraman,

I'm pleased to inform you that your manuscript has been deemed suitable for publication in PLOS One. Congratulations! Your manuscript is now being handed over to our production team.

Kind regards,

on behalf of

Dr. Saima Aleem

Academic Editor

PLOS One